# Next-Generation Sequencing of a Large Gene Panel for Outcome Prediction of Bariatric Surgery in Patients with Severe Obesity

**DOI:** 10.3390/jcm11247531

**Published:** 2022-12-19

**Authors:** Gabriele Bonetti, Kristjana Dhuli, Maria Rachele Ceccarini, Jurgen Kaftalli, Michele Samaja, Vincenza Precone, Stefano Cecchin, Paolo Enrico Maltese, Giulia Guerri, Giuseppe Marceddu, Tommaso Beccari, Barbara Aquilanti, Valeria Velluti, Giuseppina Matera, Marco Perrone, Amerigo Iaconelli, Francesca Colombo, Francesco Greco, Marco Raffaelli, Mahmut Cerkez Ergoren, Matteo Bertelli

**Affiliations:** 1MAGI’S LAB, 38068 Rovereto, Italy; 2Department of Pharmaceutical Sciences, University of Perugia, 06123 Perugia, Italy; 3MAGI EUREGIO, 39100 Bolzano, Italy; 4UOSD Medicina Bariatrica, Fondazione Policlinico Agostino Gemelli IRCCS, 00168 Rome, Italy; 5Department of Cardiology, University of Rome Tor Vergata, 00133 Rome, Italy; 6Department of Food, Environmental and Nutritional Sciences, Università degli Studi di Milano, 20122 Milan, Italy; 7Responsabile Unità di Chirurgia Bariatrica e Metabolica, Istituto Ospedaliero Fondazione Poliambulanza, 25124 Brescia, Italy; 8UOC Chirurgia Endocrina e Metabolica, Fondazione Policlinico Universitario Agostino Gemelli IRCCS, Università Cattolica del Sacro Cuore, 00168 Rome, Italy; 9Department of Medical Genetics, Faculty of Medicine, Near East University, 99138 Nicosia, Cyprus; 10MAGISNAT, Peachtree Corners, GA 30092, USA

**Keywords:** obesity, next generation sequencing, target sequencing, bariatric surgery

## Abstract

Obesity is a chronic disease in which abnormal deposition of fat threatens health, leading to diabetes, cardiovascular diseases, cancer, and other chronic illnesses. According to the WHO, 19.8% of the adult population in Italy is obese, and the prevalence is higher among men. It is important to know the predisposition of an individual to become obese and to respond to bariatric surgery, the most up-to-date treatment for severe obesity. To this purpose, we developed an NGS gene panel, comprising 72 diagnostic genes and 244 candidate genes, and we sequenced 247 adult obese Italian patients. Eleven deleterious variants in 9 diagnostic genes and 17 deleterious variants in 11 candidate genes were identified. Interestingly, mutations were found in several genes correlated to the Bardet–Biedl syndrome. Then, 25 patients were clinically followed to evaluate their response to bariatric surgery. After a 12-month follow-up, the patients that carried deleterious variants in diagnostic or candidate genes had a reduced weight loss, as compared to the other patients. The NGS-based panel, including diagnostic and candidate genes used in this study, could play a role in evaluating, diagnosing, and managing obese individuals, and may help in predicting the outcome of bariatric surgery.

## 1. Introduction

Obesity is a complex disease, with a significant genetic component, characterized by an excessive fat deposition [1,2,3,4]. It has a global prevalence of 12% and is associated with an increased risk of co-morbidities, such as metabolic syndromes, cardiovascular diseases, and cancer [5,6,7]. Individuals are defined obese when their body mass index (BMI) is over or equal to 30 [8]. Severe obesity is found in 2–6% of the world population and is diagnosed when BMI ≥35 kg/m^2^ plus at least one obesity-related comorbidity or BMI ≥ 40 kg/m^2^ [9]. Obesity is managed with two main types of therapies: nonsurgical and surgical [10]. Nonsurgical methods aim at correcting the imbalance between food intake and energy expenditure, in order to change the body composition and the metabolic status [10], while surgical methods include different bariatric surgery procedures, including laparoscopic sleeve gastrectomy, intragastric balloon, and Roux-en-Y gastric bypass [10]. Nonsurgical methods are usually ineffective in case of genetic and severe obesity; indeed, bariatric surgery is considered the most effective treatment for patients with severe obesity, while it can also be considered a last-resort treatment option in selected genetic obesity patients [10,11,12,13].

Obesity is usually due to an imbalance between energy consumption and energy expenditure, but it can also be caused by genetic, environmental, psychological and economic factors [14]. The genetic etiology of obesity is a deeply researched topic, and the genetic analysis of obese patients has shown that rare polymorphisms should also be considered for understanding molecular etiology [15,16,17,18,19]. Due to genetic heterogeneity, it can be difficult to identify genetic defects in patients with obesity. However, following clinical diagnosis, a genetic diagnosis is important for selected obesity patients, since it may provide them with personalized therapies, based on their own genetic state. Genetically, obesity can be classified as: monogenic, primarily caused by mutations in several genes involved in the leptin/melanocortin and adipogenesis pathways (such as *MC4R*, *LEP*, *LEPR*, *POMC* and *PCSK1*) [20,21,22]; syndromic, associated with neurodevelopmental abnormalities and/or other malformations due to chromosomal abnormalities or single nucleotide variations affecting genes that encode pivotal proteins in the regulation of energy balance [23]; polygenic, caused by the contribution of more than one genetic variant, whose effect is amplified in a ‘weight-gain-promoting’ environment [24]. In the last ten years, next-generation sequencing (NGS) approaches have greatly improved the rate of molecular diagnosis because of their extremely high specificity, sensitivity, accuracy, and their time- and cost-effectiveness [20,25,26,27,28,29,30]. To date, about 250 genetic variants associated with BMI or waist-to-hip ratio have been identified through genome-wide association studies and whole genome or exome sequencing [31].

A patient’s genetic risk score can be used to determine individual predisposition to obesity [32]. Moreover, in the preoperative assessment phase of bariatric surgery, genetic tests may be useful in identifying patients that will be responsive to bariatric surgery, as well as in choosing the most suitable bariatric procedure [33,34,35]. Indeed, it has been demonstrated that the variants causing monogenic obesity may be associated with reduced postoperative weight loss, especially variations in genes involved in the leptin–melanocortin pathway [36,37]. Moreover, several international guidelines on bariatric surgery intervention recognize the importance of genetics in obesity onset [38,39,40].

Here, we report the first Italian study and one of the first in Europe to evaluate the use of NGS in predicting the outcome of bariatric surgery. We performed an analysis of 316 genes by target NGS, including 72 diagnostic genes and 244 candidate genes, in 247 adult obese patients. Target sequencing of a panel of genes including diagnostic and candidate genes may be useful in evaluating, diagnosing, and managing obese individuals, and may predict the outcome of bariatric surgery.

## 2. Materials and Methods

### 2.1. Subjects and Samples

We analyzed obese Italian adults with BMI ≥ 30. All patients underwent pre-test counselling, during which clinical data—including personal and family history—were collected. The patients were informed about the significance of genetic testing. All of them gave their written informed consent, in compliance with the Declaration of Helsinki. Ethical approval and clearance were received from the Ethical Committee of Azienda Sanitaria dell’Alto Adige, Italy (Approval No. 132-2020). Genomic DNA was isolated from peripheral blood using a commercial kit (SaMag Blood DNA Extraction Kit (Sacace Biotechnologies, Como, Italy)) according to the manufacturer’s instructions.

### 2.2. Panel Design and Sequencing

We designed an NGS panel of 316 genes, comprising 72 diagnostic genes and 244 candidate genes that are linked to obesity, food intake regulation, energy homeostasis, and lipid metabolism (Appendix A). The genes included in the panel were retrieved from the Human Gene Mutation Database (HGMD Professional), Online Mendelian Inheritance in Man (OMIM), Orphanet, GeneReviews, and PubMed. The custom DNA probes were designed using Twist Bioscience technology (https://www.twistbioscience.com/ (accessed on 1 September 2022)). Genes were divided into diagnostic and candidate genes, as previously proposed by the laboratory [41]. Diagnostic genes are those that correlate to obesity, as from OMIM or scientific literature (Appendix A reports the references for the diagnostic genes-phenotype correlation [42,43,44,45,46,47,48,49,50]); candidate genes, on the other hand, correlate to obesity, adiposity, and adipocytes function in in vivo and in vitro studies. The panel included genomic targets, comprising coding exons and 15 bp flanking regions of each exon. The cumulative target length of the gene panel was 560 kb. DNA samples were processed before sequencing, as previously reported [51,52]. DNA sequencing was carried out using a MiSeq personal sequencer (Illumina, San Diego, CA, USA).

### 2.3. Bioinformatics

Fastq (forward–reverse) files were obtained after sequencing. The sequencing reads were mapped to the genome using Burrow-Wheeler Aligner (version 0.7.17-r1188) software. Duplicates were removed using SAMBAMBA (version 0.6.7) and MarkDuplicates GATK (version 4.0.0.0). The BAM alignment files generated were refined by local realignment and base quality score recalibration, using the RealignerTargetCreator and IndelRealigner GATK tools. Minor allele frequencies (MAF) were retrieved from the Genome Aggregation Database [53]. In silico prediction of the deleteriousness of nucleotide changes was performed using VarSome [54]. Each variant was classified as pathogenic, likely pathogenic, variant of unknown significance (VUS), likely benign, or benign, according to American College of Medical Genetics guidelines [55].

### 2.4. Statistics

Categorical data were expressed as absolute or as frequencies. Continuous variables were presented as mean value and standard deviation. Differences between the two groups were analyzed by using the independent samples *t*-test. The test was two tailed and a *p* value < 0.05 was considered as statistically significant.

## 3. Results

We enrolled 247 obese patients (73% female and 27% male), with a median age of 48 ± 11 years, who were analyzed with a NGS panel of 316 genes, comprising 72 diagnostic genes and 244 candidate genes. Twelve patients carried deleterious variants in diagnostic genes. Another 21 patients carried at least one deleterious variant in candidate genes. In particular, 11 heterozygous deleterious variants in 9 diagnostic genes (BBS1, BBS2, BBS5, BBS9, C8orf37, CEP290, MC4R, MCHR1, MKS1), and 17 heterozygous deleterious variants in 11 candidate genes (APOE, DNAAF1, ESR1, GHR, GUCY2C, NCOA2, NPC1, PDX1, RYR1, STRA6, and ZNF423) were detected. The genetic variants identified in the diagnostic genes are reported in Appendix A. Those identified in the candidate genes are reported in Appendix A. Finally, the list of diagnostic and candidate genes for which genetic variants were identified are reported in Appendix A. Moreover, the Appendix A report the preliminary molecular dynamics simulations that were carried out to study the effect of deleterious genetic variants [56,57,58,59,60,61,62,63].

Following the genetic analysis, 25 patients agreed to undergo a follow-up of 6 and 12 months to evaluate their weight loss after bariatric surgery. All the 25 patients completed the follow-up. Patients were subsequently divided in two groups, based on the pathogenicity classification of the genetic variant identified by the NGS sequencing: Non-deleterious (ND; patients in which only benign, likely benign or variants of uncertain significance were identified) and Deleterious (D; patients in which likely pathogenic variants were identified, Appendix A). The patients underwent different types of bariatric surgery: among the ND group, 5/20 underwent sleeve gastrectomy, 3/20 underwent SADI-S, 11/20 underwent gastric bypass and 1/20 underwent endoscopic sleeve gastroplasty; among the D group, 2/5 underwent sleeve gastrectomy and 3/5 underwent gastric bypass. Percentage of weight loss (%WL), calculated as (weight loss/weight pre-surgery) × 100, was evaluated for the two groups (Figure 1). Weight loss and pre-surgery clinical data are presented in Table 1. As it can be seen from Figure 1, at 6 months, patients of the two groups had a similar %WL (28.1% for ND and 25.7% for D), while at 12 months, the %WL of ND patients was higher (38.4%) than the %WL of D patients (23.7%). The %WL of the 25 patients considered altogether was 35.7%. Moreover, the type of bariatric surgery did not affect the %WL. Indeed, considering the two most represented types of bariatric surgery, the %WL of patients that underwent sleeve gastrectomy (27.4%) was not statistically different compared to the %WL of patients that underwent gastric bypass (26.4%). Finally, at 12 months, none of the patients had a percentage of excess weight loss (%EWL) lower than 50%.

## 4. Discussion

The global rise in obesity contributes to a significant number of diseases with high morbidity [20,27,28]. Indeed, obesity increases the risk of cardiovascular and metabolic diseases, resulting in a huge social impact and economic cost for national health systems [64]. Obesity can be caused by genetic mutations, a kind in which pharmacological and non-surgical treatments are usually not available or poorly efficient [10]. Currently, bariatric surgery is the most effective treatment for patients with severe obesity [65], and it is now clear that several single nucleotide polymorphisms may be associated with poor response to it [66,67]. However, at the moment there are no reliable biomarkers for evaluating individual responses to bariatric surgery.

In this study, we sequenced the genome of 247 patients with severe obesity before they underwent bariatric surgery. Among them, 25 were evaluated for their weight loss at 6 and 12 months after surgery and were divided in two groups. Considering that the weight and BMI of the two groups were not similar, we evaluated the %WL instead of the absolute weight loss. As can be seen from Figure 1, at 6 months, patients of the two groups had a similar %WL, which was slightly higher for ND patients. At 12 months, however, %WL of D patients was slightly reduced, while %WL of ND patients increased: thus, at 12 months, the difference in %WL between the two groups was much higher and statistically significant (*p* < 0.05), suggesting a role of genetic variants in bariatric surgery outcome [68].

Moreover, many of the diagnostic genes that were found mutated are correlated to the Bardet–Biedel Syndrome (BBS). BBS is an autosomal recessive syndrome that is characterized by several symptoms, among which are intellectual disability and obesity [69]. Among the eleven variants identified in diagnostic genes, eight deleterious mutations in nine patients were in genes related to BBS: BBS5, BBS2, BBS9, BBS1, C8orf37, CEP290, and MKS1 (Appendix A). Thus, remembering that clinical observations are needed as a first step to distinguish between syndromic and non-syndromic obesity, we propose the genetic screening of patients with severe obesity as a useful tool to be used systematically to identify underlying genetic causes. Moreover, a typical symptom of ciliopathies such as BBS is hyperphagia [70], and all the patients of the LP group were hyperphagic. The outcome of bariatric surgery in hyperphagic patients is variable and less efficacious [71]. Therefore, genetic screening of patients with severe obesity and hyperphagia could be critical in evaluating the best therapeutic procedure.

All the genes in which likely pathogenic variants were identified are correlated to the onset of obesity [72,73,74,75,76,77]. Thus, we decided to perform preliminary molecular dynamics simulations, in order to functionally characterize some of the genetic variants identified. As it was reported in the Appendix A, the molecular dynamics studies on APOE, BBS9 and GHR suggested that the variants partly disrupted the structure of the proteins of interest. Indeed, all of them replaced amino acids of different physicochemical properties. In APOE, proline at 102 was replaced by arginine, in GHR, tyrosine at 240 was replaced by histidine, and in BBS9, serine at 88 was replaced by leucine. The replacement of proline by arginine (APOE) gave rise to local fluctuations of the structure (indicated by the RMSF plots), while the replacement of tyrosine by histidine, and serine by leucine in GHR and BBS9, respectively, disrupted the hydrogen bonding network in the neighboring residues. Molecular dynamic studies were not carried out for ZDF423 and PDX1 variants because only predicted structures were available online and the variants resided in positions with low prediction confidence.

Our results support already published scientific articles [36,37], and indicate that genetic testing could be useful to predict the outcome of bariatric surgery, and that patients with predicted deleterious variants to genes correlated to obesity could respond less to bariatric surgery.

The major limitation of this study is the low number of patients recruited for the assessment of weight loss. Nevertheless, this is the first Italian study and one of the first in Europe that evaluates the connection between genetic obesity and bariatric surgery outcomes. Moreover, a variable that should be taken into account is the type of bariatric surgery that was performed on each patient. In considering these limitations, our results could spur newest studies with higher cohorts of patients, which will foster the role of genetics in affecting the outcome of bariatric surgery.

## 5. Conclusions

We genetically screened 247 severe obese patients and we clinically followed 25 of them to assess their weight loss after bariatric surgery, the most up-to-date treatment for severe obesity. In the meantime, we performed in silico modelling studies of the likely pathogenic variants, which supported their deleterious effect on the protein structure. It resulted that patients carrying likely pathogenic variants had a reduced weight loss. Considering that more research on a bigger cohort and with similar types of bariatric surgery is needed to support our findings, we propose that testing the genetics of severe obesity patients before performing bariatric surgery could be useful in the clinical management of patients with severe obesity.

## Figures and Tables

**Figure 1 jcm-11-07531-f001:**
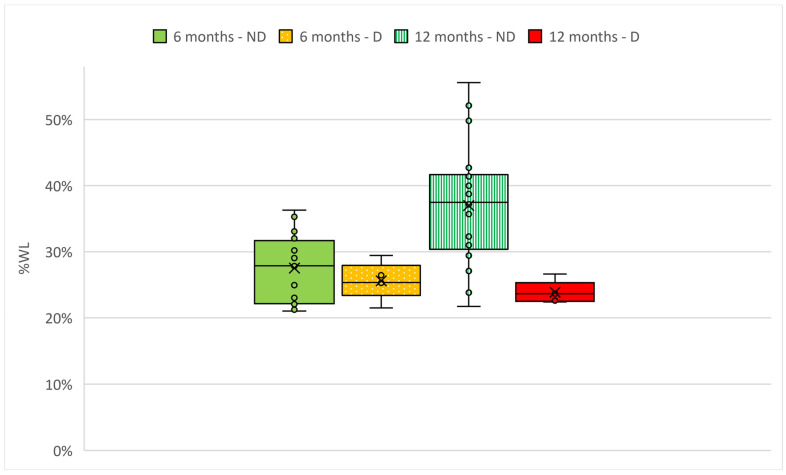
Percentage of weight loss (%WL) after bariatric surgery in two groups of adults with obesity. Patients were divided in two groups ND (Non-deleterious) and D (Deleterious), based on the pathogenicity classification of the genetic variant identified with the NGS sequencing. Data were analyzed with independent samples *t*-test analysis, with *p* ≤ 0.05.

**Table 1 jcm-11-07531-t001:** Weight loss of bariatric surgery patients after 6 and 12 months since bariatric surgery. Pre-surgery clinical data on weight, BMI, hyperphagia, glycemia and total cholesterol are reported. Patients were divided in two groups, based on the pathogenicity classification of the genetic variant identified with the NGS sequencing. Data were analyzed with independent samples *t*-test analysis.

Group	Non-Deleterious	Deleterious	*p*-Value
N of patients	20	5	-
Females/Males	15/5	4/1	-
BMI	45.0 ± 7.5	36.5 ± 3.5	<0.05
Age	50.4 ± 10.7	49.4 ± 12.6	>0.05
Height (m)	1.7 ± 0.1	1.8 ± 0.2	>0.05
Glycemia (mg/dL)	96.5 ± 21.1	91.5 ± 26.8	>0.05
Total cholesterol (mg/dL)	188.2 ± 36.5	119.3 ± 42.4	<0.05
Pre-surgery weight (kg)	129.4 ± 25.4	118.6 ± 19.3	>0.05
Hyperphagia (%)	50.0	100.0	-
BMI after 6 months	31.9 ± 4.0	27.2 ± 15.0	<0.05
Weight loss after 6 months (kg)	36.4 ± 13.3	30.5 ± 6.2	>0.05
BMI after 12 months	28.4 ± 2.8	28.4 ± 2.9	>0.05
Weight loss after 12 months (kg)	49.7 ± 19.7	28.2 ± 3.8	<0.05
%EWL after 12 months (%)	81.6 ± 10.3	84.3% ± 12.3	>0.05

## Data Availability

Data are contained within the Appendix A.

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
