# Peer review of "Next-Generation Sequencing of a Large Gene Panel for Outcome Prediction of Bariatric Surgery in Patients with Severe Obesity"

_jcm, 2022, doi:10.3390/jcm11247531_

Round 1
Reviewer 1 Report
Dear editor,
I have reviewed the work by Bonetti G., et entitled: Next-Generation Sequencing of a large gene panel for outcome prediction of bariatric surgery in patients with severe obesity, in which they analyze genes related to deleterious and non-deleteric forms in obese patients, for means of next generation sequence. The work provides new information on the genetics of obesity, and adds new response predictors for this group of patients. The work is well developed and the objectives are corroborated by its results. The discussion is analyzed in an adequate way.
Author Response
Please see attachment.
Dear reviewers,
thank you very much for all your kind reviews. We will answer to all your kind comments and suggestions in this word file.
Best regards,
Gabriele Bonetti
Reviewer 1
Dear editor,
I have reviewed the work by Bonetti G., et entitled: Next-Generation Sequencing of a large gene panel for outcome prediction of bariatric surgery in patients with severe obesity, in which they analyze genes related to deleterious and non-deleteric forms in obese patients, for means of next generation sequence. The work provides new information on the genetics of obesity, and adds new response predictors for this group of patients. The work is well developed and the objectives are corroborated by its results. The discussion is analyzed in an adequate way.
Dear reviewer,
Thank you very much for your kind review.
Best regards,
Gabriele Bonetti
Reviewer 2
Dear reviewer,
Thank you very much for your kind review. We will answer to all your points below.
Best regards,
Gabriele Bonetti
The authors aimed to evaluate the use of NGS in predicting the outcome of bariatric surgery. This is an interesting topic which has also been covered in a very recent publication of Campos et al in Obesity Surgery (2022) (https://doi.org/10.1007/s11695-022-06122-9). (The authors should consider to read and cite this article)
Thank you for the suggested article. We found it very interesting and we added it as a reference (line 87).
However, in the present study major revisions are needed to support the interpretations made by the authors and pivotal information are missing.
Very Important characteristics of the Patients are missing:
-age of the patients,
-type of surgery,
- list of the variants of the 5 patients with deleterious variant,
- status of the variant (heterozygous or homozygous )
- height of the patients (according to the BMI and weight the average height of the 5 patients with deleterious variants would be 1,8 meter, which seems very high for a group of 4 women and 1 man…)
- …./…
We added the suggested information:
- in Table 1, the age and the height of the patients of the two groups
- in lines 157-161, the type of surgery
- in lines 143-144, the status of the variants
- in Table S6, the list of variants of the 5 patients with deleterious variants
Without those informations no hypothesis or conclusion can be made to explain why these 5 patients lost less weight 12 months after surgery than the other 20. There are too many confounding factors. And, as mentioned by the authors, the small number of patients does not allow to draw a conclusion either, especially if there is great heterogeneity between patients, which is very likely the case (type of surgery, ...), and the difference of the subgroup size does not help. I also have doubts about the relevance of using a t test (parametric) to highlight differences on such a small number of patients and more with a comparison during a kinetics...
We are aware of the difficulty of making a definite conclusion. Thus, we modified the conclusions to underline that more research on a bigger cohort is needed to support our findings (line 246-248). Moreover, in the discussion we have already written that “Also considering these limitations, our results could spur newest studies with higher cohorts of patients, which will foster the role of genetics in affecting the outcome of bariatric surgery”, being aware of the limitations of the study. Nevertheless, also considering already published articles that underline the correlation between genetics and bariatric surgery outcome, and that our manuscript would be the first published study in an Italian population to prove this correlation, we think that our results could be important in fostering new research in this field.
t test has already been used to test differences in very small groups (Winter, J.C.F. (2013) "Using the Student's t-test with extremely small sample sizes," Practical Assessment, Research, and Evaluation: Vol. 18, Article 10). Moreover, the difference in the subgroup sizes should not be a problem, considering that a bigger control group could increase the power of the test (Introduction to the theory of statistics: Mood A.M., Graybill F.A., Boes D.C).
The difference of the pre-operative weight / BMI between the 2 subgroups could also be a confounding factor for the interpretation.
Considering that the weight and BMI of the two groups were not similar, we evaluated the %WL instead of the absolute weight loss. Evaluating the %WL should reduce the confounding factor.
The authors give a list of 244 "candidate" genes and 72 "diagnostic" genes, but it is not clear which pathology the authors consider these genes to be "candidate" or "diagnostic"?
We modified table S1 in order to add the pathologies (Gene-phenotype relationship) correlated to diagnostic genes. Moreover, the pathologies/phenotypes correlated to the candidate genes in which deleterious variants were identified were already present in Table S5 (Gene-phenotype relationship).
Beyond the conclusions to be reviewed, adjustments should be made in the text, see below some examples
IN the introduction:
line 56-58 : it not reasonnable to write that bariatric surgery is the most effective treatment for genetic obesity
Thank you for the important suggestion. We modified as: “Nonsurgical methods are usually ineffective in case of severe obesity; indeed, bariatric surgery is considered the most effective treatment for patients with severe obesity, while is can also be considered a last-resort treatment option in selected genetic obesity patients” (lines 55-59).
line 64-66 : "However, a genetic diagnosis is important for these patients,"... Which patients ? all obese patients ? It doesn't seem reasonable to test patients without establishing criteria (such as age of obesity, BMI, hyperphagia or not etc…)
Thank you for the comment. We agreed that the sentence was not clear and that clinical observation is very important, thus we modified as “However, following clinical diagnosis, a genetic diagnosis is important for selected obesity patients, since it may provide them with personalized therapies, based on their own genetic state.” (lines 65-67)
line 66-68 : “Genetically, obesity can be classified as: monogenic, primarily caused by mutations in genes involved in the leptin/melanocortin and adipo-67 genesis pathways (MC4R, LEP, and LEPR) [18];”
there are other genes : POMC, PCSK1, SIM1, MRAP2 ADCY3…. , but also SH2B1, etc… Also it exists more recent publications, with the up-dates list of genes
We added more recent publications with an up-to-date list of genes (lines 69-70).
In the discussion:
Line 173-175 : “Obesity can be caused by genetic mutations, a kind that is generally treated via phar-173 macological and non-surgical treatments that are usually not available or poorly efficient 174 [10].” In contradiction to what was stated in the introduction line 56-58 (see comment above)
We modified both sentences, and now they should not be in contradiction. Indeed, we wrote:
- “Nonsurgical methods are usually ineffective in case of genetic and severe obesity; indeed, bariatric surgery is considered the most effective treatment for patients with severe obesity, while is can also be considered a last-resort treatment option in selected genetic obesity patients” (lines 55-59)
- “Obesity can be caused by genetic mutations, a kind in which pharmacological and non-surgical treatments are usually not available or poorly efficient” (lines 181-183).
Line 187-189 : “Moreover, the overall %WL was higher than the reported 187 values in literature, while the %WL of D patients was lower. Indeed, in a recent scientific 188 article a group of 50 bariatric surgery patients have lost 30.2% of their weight one year 189 after surgery [54]”, Is it really comparable ? was it the same type of surgery
Thank you very much for your suggestion. We agreed that to remove the sentence, because the results were not fully comparable.
Line 187-189 : “Thus, the genetic screening of patients with severe obesity should be 197 used systematically to distinguish between syndromic and non-syndromic obesity”. Really I don’t think you can really conclude that from the sentence above (line 192 to 197)… Clinical observations are neede as a first step to distinguish between syndromic and non syndromic obesity
We agreed that clinical observation is very important as a first step to distinguish between syndromic and non-syndromic obesity. Therefore, we modified the sentence as follows: “Thus, remembering that clinical observations are needed as a first step to distinguish between syndromic and non-syndromic obesity, we propose the genetic screening of patients with severe obesity as a useful tool to be used systematically to identify underlying genetic causes.” (lines 207-210)
Reviewer 3
The study reported the use of NGS in predicting the outcome of bariatric surgery. They performed an analysis of 316 genes by target NGS, including 72 diagnostic genes and 244 candidate genes, in 247 adult obese patients, and identified 11 deleterious variants in 9 diagnostic genes and 17 deleterious variants in 11 candidate genes. Further, they proved that these genes may predict the outcome of bariatric surgery by follow up 25 patients.
- Besides body weight, is there any data on other metabolic indicators, as serum glucose, lipid and HbA1c? Body fat content and lean mass?
Dear reviewer, thank you very much for your kind comments. We added to Table 1 the age and the height of the patients, in lines 157-161 the type of surgery, in lines 143-144 the status of the variants, in Table S1 the Gene-phenotype correlation of diagnostic genes, and in Table S6 the specific deleterious variants identified in the clinically followed patients. Unfortunately, no other clinical data can be added.
- Only 25 patients underwent a follow-up of 6 and 12 months to evaluate weight loss after bariatric surgery, if possible, increase the sample size.
We know that the sample size was not big, but it cannot be increase. Therefore, also considering other reviews, we decided to modify the conclusions (lines 246-248). Nevertheless, also considering that the correlation between genetics and efficacy of bariatric surgery was already suggested in other studies, we think that this study could be important to spur newest studies with higher cohorts of patients.
Best regards,
Gabriele Bonetti

Reviewer 2 Report
The authors aimed to evaluate the use of NGS in predicting the outcome of bariatric surgery. This is an interesting topic which has also been covered in a very recent publication of Campos et al in Obesity Surgery (2022) (https://doi.org/10.1007/s11695-022-06122-9). (The authors should consider to read and cite this article)
However, in the present study major revisions are needed to support the interpretations made by the authors and pivotal information are missing.
Very Important characteristics of the Patients are missing:
-age of the patients,
-type of surgery,
- list of the variants of the 5 patients with deleterious variant,
- status of the variant (heterozygous or homozygous )
- height of the patients (according to the BMI and weight the average height of the 5 patients with deleterious variants would be 1,8 meter, which seems very high for a group of 4 women and 1 man…)
- …./…
Without those informations no hypothesis or conclusion can be made to explain why these 5 patients lost less weight 12 months after surgery than the other 20. There are too many confounding factors. And, as mentioned by the authors, the small number of patients does not allow to draw a conclusion either, especially if there is great heterogeneity between patients, which is very likely the case (type of surgery, ...), and the difference of the subgroup size does not help. I also have doubts about the relevance of using a t test (parametric) to highlight differences on such a small number of patients and more with a comparison during a kinetics...
The difference of the pre-operative weight / BMI between the 2 subgroups could also be a confounding factor for the interpretation.
The authors give a list of 244 "candidate" genes and 72 "diagnostic" genes, but it is not clear which pathology the authors consider these genes to be "candidate" or "diagnostic"?
Beyond the conclusions to be reviewed, adjustments should be made in the text, see below some examples
IN the introduction:
line 56-58 : it not reasonnable to write that bariatric surgery is the most effective treatment for genetic obesity
line 64-66 : "However, a genetic diagnosis is important for these patients,"... Which patients ? all obese patients ? It doesn't seem reasonable to test patients without establishing criteria (such as age of obesity, BMI, hyperphagia or not etc…)
line 66-68 : “Genetically, obesity can be classified as: monogenic, primarily caused by mutations in genes involved in the leptin/melanocortin and adipo-67 genesis pathways (MC4R, LEP, and LEPR) [18];”
there are other genes : POMC, PCSK1, SIM1, MRAP2 ADCY3…. , but also SH2B1, etc… Also it exists more recent publications, with the up-dates list of genes
In the discussion:
Line 173-175 : “Obesity can be caused by genetic mutations, a kind that is generally treated via phar-173 macological and non-surgical treatments that are usually not available or poorly efficient 174 [10].” In contradiction to what was stated in the introduction line 56-58 (see comment above)
Line 187-189 : “Moreover, the overall %WL was higher than the reported 187 values in literature, while the %WL of D patients was lower. Indeed, in a recent scientific 188 article a group of 50 bariatric surgery patients have lost 30.2% of their weight one year 189 after surgery [54]”, Is it really comparable ? was it the same type of surgery
Line 187-189 : “Thus, the genetic screening of patients with severe obesity should be 197 used systematically to distinguish between syndromic and non-syndromic obesity”. Really I don’t think you can really conclude that from the sentence above (line 192 to 197)… Clinical observations are neede as a first step to distinguish between syndromic and non syndromic obesity…
Author Response

(The authors gave the same response as above.)

Reviewer 3 Report
The study reported the use of NGS in predicting the outcome of bariatric surgery. They performed an analysis of 316 genes by target NGS, including 72 diagnostic genes and 244 candidate genes, in 247 adult obese patients, and identified 11 deleterious variants in 9 diagnostic genes and 17 deleterious variants in 11 candidate genes. Further, they proved that these genes may predict the outcome of bariatric surgery by follow up 25 patients.
1. Besides body weight, is there any data on other metabolic indicators, as serum glucose, lipid and HbA1c? Body fat content and lean mass?
2. Only 25 patients underwent a follow-up of 6 and 12 months to evaluate weight loss after bariatric surgery, if possible, increase the sample size.
Author Response

(The authors gave the same response as above.)
